# Commingling of Waste Rock and Tailings to Improve "Dry Stack" Performance: Design and Evaluation of Mixtures

**Ralph Burden** [1,*] **and G. Ward Wilson** [2]

1    Frontera Geotechnical, Squamish, BC V8B 0K8, Canada
2    Department of Civil & Environmental Engineering, University of Alberta, Edmonton, AB T6G 2W2, Canada
*    Correspondence: ralph@fronterageo.ca

**Abstract:** Mine tailings are typically deposited as slurry and stored in impoundment dams. These structures pose a serious geotechnical risk, and are difficult to successfully reclaim at the end of mining. An increasingly common alternative to traditional tailings disposal is "dry stacking": the placement of tailings dewatered using filtration in a self-supporting stack. It has been demonstrated that the addition of rock (termed "commingling") to a filtered tailings stack has the potential to improve the geotechnical performance of the stack and may make large-scale dry stacking more economically viable. This paper discusses the application of commingling to tailings dry stacking, specifically relating to the design and evaluation of commingled blends of waste rock and tailings. The authors present a review of existing mix design theory, and present an extended theoretical model to predict the structure and behavior of blends of waste rock and tailings, based upon mix ratio and density. This paper is based upon established theory, but is extended to consider the case of loosely placed materials and the effect of volume change on structural configuration. The extended model may be used to describe the geotechnical behavior of commingled filtered tailings and waste rock. It is postulated that the geotechnical behavior of blends, and the primary mechanism of volume change, is governed by particle configuration. A brief discussion of experimental methods to evaluate the structure and configuration of commingled mine wastes is also presented.

**Keywords:** filtered tailings; commingling; mix design; dry stacking

## 1. Introduction

Conventionally, mine tailings are discharged and transported as a slurry and stored behind dams, which are often constructed from the tailings themselves [1]. Tailings dams can present a challenge for reclamation and closure, and are a long-term geotechnical risk and liability. Several large, high-profile and sometimes tragic failures have occurred in recent years, such as Mount Polley [2], Fundão [3] and Feijão [4], the latter of which resulted in the loss of over 250 lives. In response, traditional tailings dams are increasingly becoming seen as unacceptable by many stakeholders, and tailings management practices are now coming under increasing scrutiny by major investors.

The global appetite to reduce risk associated with tailings management, combined with advances in filtration technology, has led to a resurgence of interest in recent years in "dry stacking", the deposition of tailings in a self-supporting pile that has the potential to eliminate or reduce the need for a dam. Dry stacking is typically achieved by using pressure or vacuum filtration to rapidly dewater tailings (usually to around 80% solids), which are then transported by truck or conveyor and then spread and compacted. While dry stacking is now considered for most major projects, it has thus far only been applied at small to medium-sized mines; the largest currently operating mines have throughputs of around 30,000 tons per day [5,6]. A potential barrier to the successful scale-up of filtered tailings is the cost and operation complexity involved in the placement and compaction of filtered tailings using conventional construction equipment.

Commingling of waste rock and tailings may offer a solution to this problem. The Merriam-Webster dictionary defines the word "commingle" as to "blend thoroughly into a harmonious whole". The word commingled has been in the English vocabulary perhaps since Shakespeare used it in his play *Hamlet*. Commingled, or blended, waste rock and tailings have been the subject of numerous laboratory [7–13] and field-scale [14] trials over the years that have demonstrated that this material can be engineered to have preferential geotechnical or geochemical properties compared to tailings or waste rock alone. The addition of waste rock to a filtered tailings stack during transit, using the same conveyor system used to transport the material to create a blended material, has the potential to offer significant advantages over established dry stacking methods, which may make large-scale dry stacking more economically viable.

This paper discusses the application of commingling to tailings dry stacking, specifically relating to the design and evaluation of commingled blends of waste rock and tailings. A brief, high-level review is presented of the current state of practice in tailings dry stacking and commingling. A review of existing mix design theory is presented, and its applicability to dry stacking applications is discussed. An extended theoretical model is presented for the case of commingled filtered tailings and waste rock. In addition, the paper includes a brief discussion of experimental methods that may be used to evaluate the structure and configuration of commingled mine wastes.

## 2. Background

### 2.1. Filtered Tailings "Dry Stacking"

Densified tailings technologies are broadly divided into three categories based on the mechanical properties and method of dewatering: thickened tailings (typically ~50%–65% solids by mass), paste tailings (typically 70%–80% solids) and filtered tailings (>80% solids). Figure 1 summarizes the properties, and pros and cons, of different densified tailings technologies.

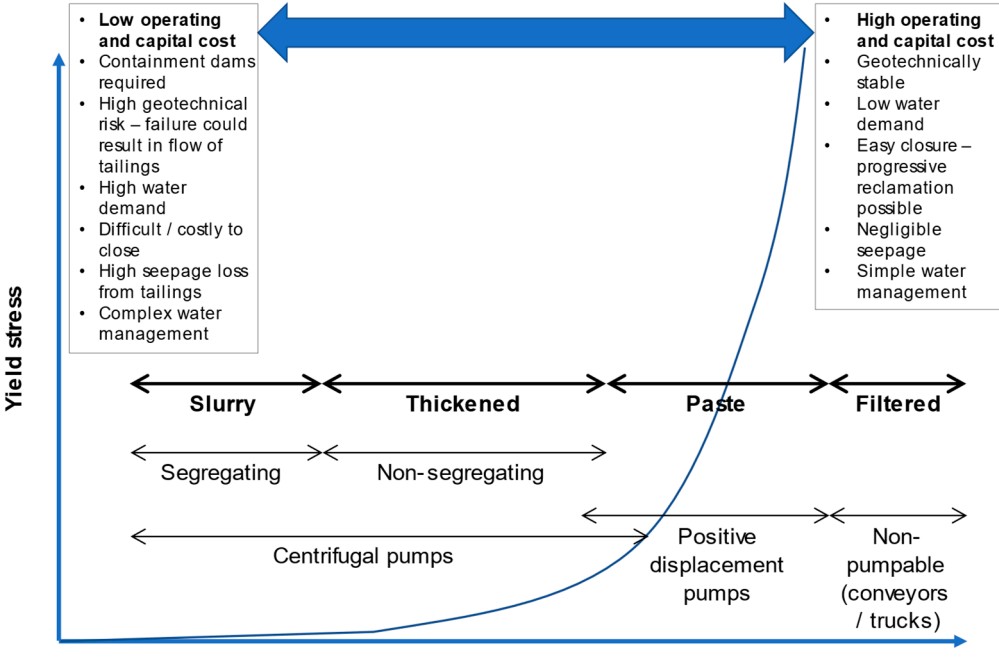

**Figure 1.** The tailings continuum, after Davies and Rice (2001) [15] and Jewell and Fourie (2006) [16].

Thickened and paste tailings technologies have many potential benefits, namely reduced water consumption, reduced waste volumes and improved geotechnical stability, whilst maintaining relatively low operating costs and good geochemical

performance [16,17]. However, some form of containment structure is still required for the deposition of thickened tailings or paste tailings alone, as well as uncompacted filtered tailings. Construction of a stable, self-supporting "dry stack" requires both filtration and compaction or co-disposal technologies.

Filtration is widely used in many fields to separate solids and liquids. The material is placed on a porous medium (such as filter paper or cloth), and a pressure gradient is applied across the medium, causing flow of filtrate through the medium and leaving behind a densified "cake". In tailings applications, the pressure gradient may be applied by vacuum, mechanical squeezing action or via the feed pump. The most commonly used filter plant configurations are drums, stacked plates in either horizontal or vertical alignment, or horizontal belts. The tailings are typically run though a conventional thickener prior to filtration. The filter cake is often left to dry to the optimum moisture content, or slightly dry of optimum, before compaction into the stack. The efficiency of the process is very much dependent upon the properties of the tailings feed. Low fines content, low clay content and high specific gravity are all preferable for filtration [18].

The first published use of filtered tailings technology is in the coal mining industry [18]. Coal tailings, produced in the washing of coal, have a high fines content and low permeability, which makes dewatering by more conventional means difficult. As of 2008, there were reported to be 37 operating mines worldwide employing filtered tailings technologies, with production rates of up to 24,000 t/day [19]. The number of operating facilities has undoubtedly increased in recent years, although no comprehensive list exists in the literature. Crystal and Hore [5] prepared an inventory of 19 currently operating dry stacks, although almost all of them are small operations below 10,000 t/day. The largest currently operational mine employing filtered tailings that has published a case study is Karara [6] in Western Australia, a 30,000 t/day magnetite mine. Tailings at this facility are pressure filtered using vertical plates to around 85% solids, whereupon they are transported using an overland conveyor system and stacked in a radial pattern using conveyors and stackers. Recent technological advances led to the development of very high volume filter plates with lower maintenance requirements, which may assist in making filtration viable for large-scale mining [20].

The published literature on the geotechnical properties of filtered tailings is sparse and generally restricted to case histories [21–24]. Generally speaking, filtered metal mine tailings have been shown to have similar behavior to unsaturated silty sand, which is amenable to compaction and stacking.

### 2.2. Co-Disposal of Mine Wastes

The term "co-disposal" generally refers to the disposal of different waste streams in the same facility. Co-disposal can take on many forms with varying degrees of mixing of the wastes. Figure 2 shows common forms of co-disposal, ordered by degree of mixing.

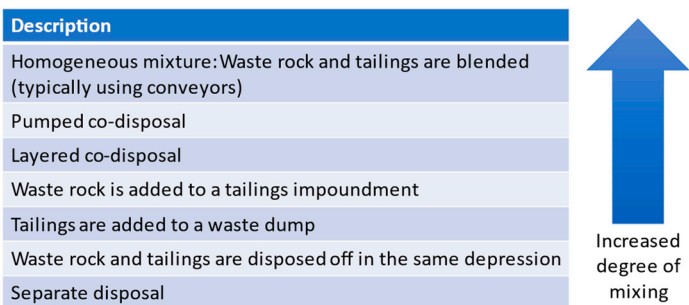

**Figure 2.** Forms of co-disposal, after Wickland (2006) [7].

The earliest use of co-disposal is probably in underground backfills, often combined with a binder such as Portland cement [25]. Pumped co-disposal has also been implemented successfully at several coal mines around the world, and has been shown to reduce waste

volumes and the need for containment [26]. Other approaches that have been demonstrated include layered co-disposal, or "commingling", which typically involves the placement of fine tailings layers inside a waste rock pile to act as a barrier to seepage or oxygen flux, and waste rock "inclusions" which are waste rock dykes placed inside tailings ponds as the ponds are raised to promote drainage and improve overall stability and seismic resistance. A summary of these techniques is given by Bussiere [27]. The focus of this paper is on blended, or "commingled", co-disposal.

### 2.3. Commingling of Tailings and Waste Rock

Numerous laboratory [7–13] and field-scale [14] trials focusing on blended waste rock and tailings have demonstrated that these materials can have favorable geotechnical and geochemical properties when compared to tailings and waste rock alone, combining the high shear strength and comparatively low compressibility of the waste rock with the permeability and water retention properties of the tailings, and potentially allowing tailings stacks to remain unsaturated. Field trials have demonstrated that waste rock and filtered tailings can successfully be blended using conveyors [28]. Field trials in the Canadian oil sands have demonstrated the successful stacking of very-clay-rich oil sands tailings, which are widely regarded as being challenging to dewater by conventional methods, by commingling with clay shale overburden using conveyors [29,30].

## 3. Mixture Design Theory for Commingled Waste Rock and Tailings

### 3.1. Introduction

Wickland et al. [7] presented a theoretical basis for the design of "paste rock" blends, based on particle structure, as well as methods for predicting blend structure and properties from the mix ratio. The optimum mix ratio was found to be the "just-filled" point, where the dewatered tailings fill the voids between the waste rock particles. Figure 3 shows structural configurations of "paste rock"-style blends.

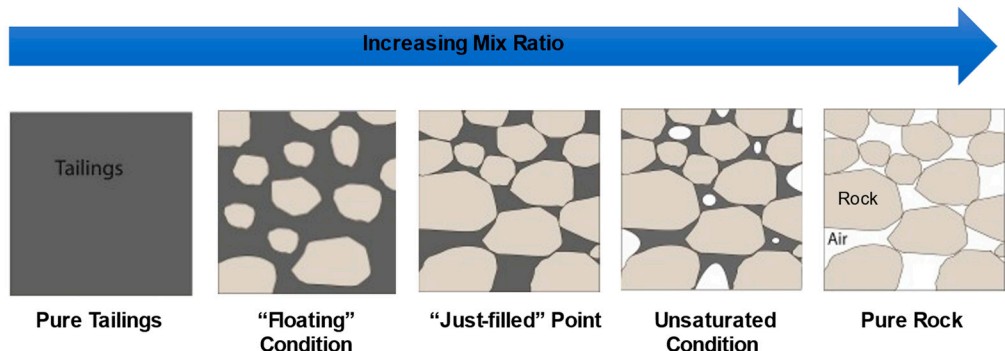

**Figure 3.** Blend configurations after Wickland et al. (2006) [7].

Figure 4 shows the general relationship between the density and mix ratio for commingled tailings and rock. The "optimum" mix ratio, which forms the densest possible blend, is a function of the tailings density, the waste rock skeleton porosity and the waste rock moisture content. Increasing rock porosity and tailings density results in an optimum mix ratio with lower rock content.

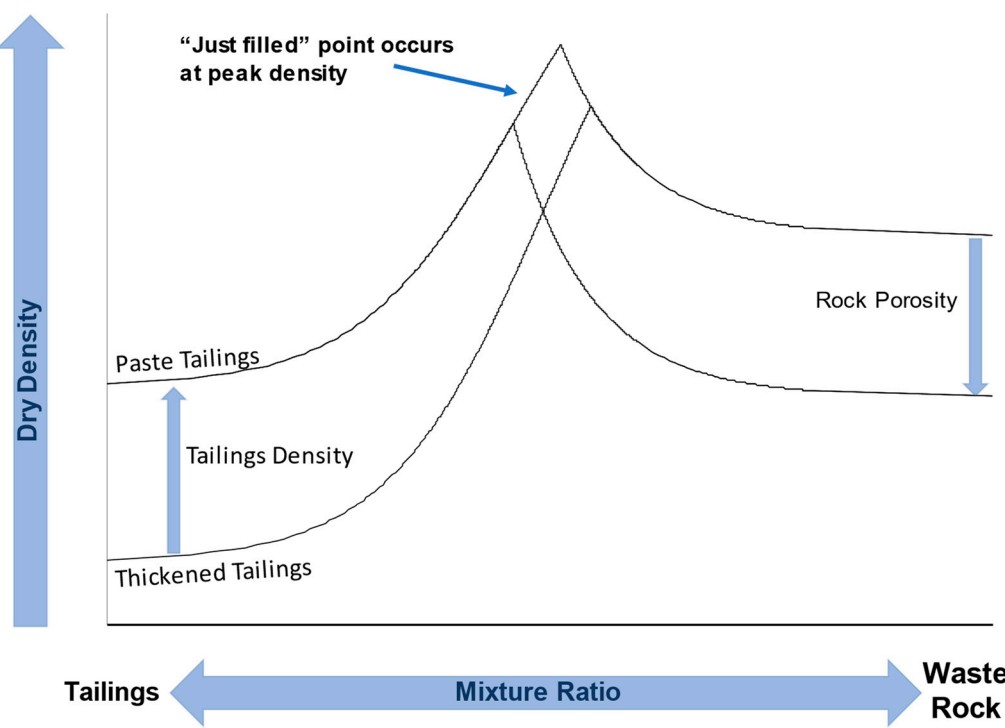

**Figure 4.** Mix design theory after Wickland et al. (2006) [7].

### 3.2. Ternary Diagrams

Ternary diagrams constructed using the approach developed by Charles and Charles [31] to characterize three-phase mixtures are widely used in the oil sands industry [32] to describe the structure and behavior of composite tailings, which contain both a fines and a sand component. The same approach provides a useful framework to describe the structure and behavior of waste rock–tailings blends, with respect to mass relations. Any theoretical combination of rock solids, tailings solids and water can be represented by a point inside the triangle. The ternary diagram approach provides a simple and effective tool for quickly estimating the bulk properties of a blend of a given tailings solids content and mix ratio, which may assist in high-level tailings planning and conceptual design.

Figure 5 shows the construction of the ternary diagram. The apices of the triangle represent 100% by mass water (top), rock solids (bottom left) and tailings solids (bottom right). Points inside the triangle represent any theoretical blend of tailings solids, rock solids and water. Points along the axis connecting 100% water and 100% tailings apices represent tailings only. Straight lines drawn from this line to the 100% rock apex (shown in magenta on Figure 5) represent blends with constant tailings solids content. Straight lines drawn from the bottom axis to the 100% water apex represent constant mix ratios. It should be noted that unlike ternary diagrams used in the oil sands industry, which consider coarse solids and fine solids as defined by a specific particle size, the diagrams discussed in this paper consider rock solids and tailings solids as defined by origin.

Figure 6 shows the ternary diagram annotated with general material property and blend configuration boundaries. The figure was generated using a waste rock moisture content of zero and specific gravity of tailings and waste rock solids of 2.7. The solid black lines show the "just-filled" point, or optimum mix ratio described by Wickland et al. [7], for waste rock skeleton porosities of 40% and 50%, calculated based upon the assumption that the tailings are fully saturated at the "just-filled" point. These lines define the division between blends whose behavior is dominated by fines (blends in the "floating" condition) and blends whose behavior is rock-dominated (blends with a continuous phase of rock particles in contact). The dashed lines show assumed tailings property boundaries based on solids content. For consistency with other researchers, the mix ratio $R$ is defined as dry

mass of waste rock solids/dry mass of tailings solids. Figure 7 shows the ternary diagram annotated with typical solids contents of different tailings dewatering technologies.

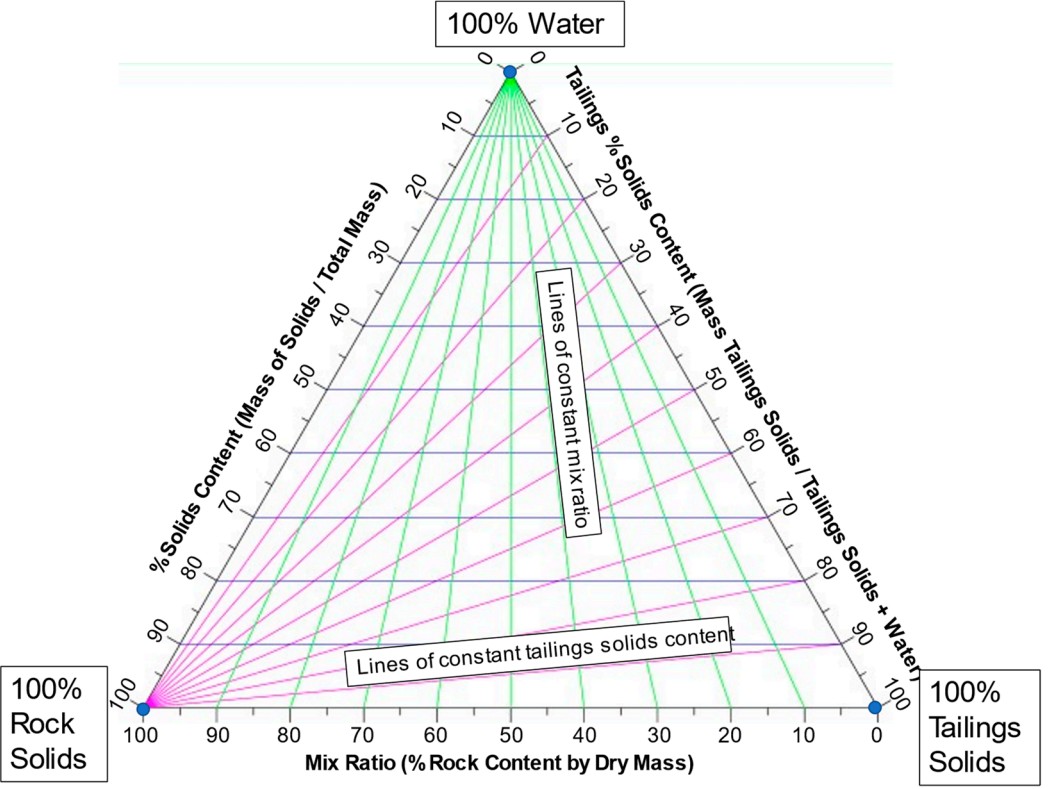

**Figure 5.** Ternary diagram for waste rock–fine tailings blends.

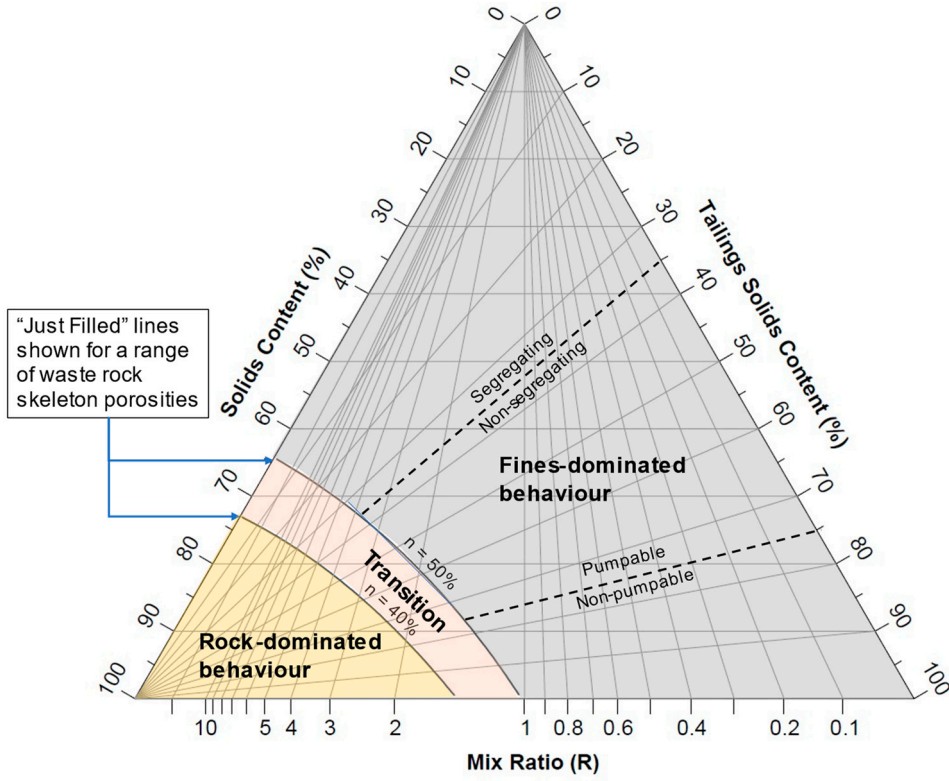

**Figure 6.** Ternary diagram showing blend configuration vs. mix ratio for fine tailings–waste rock blends.

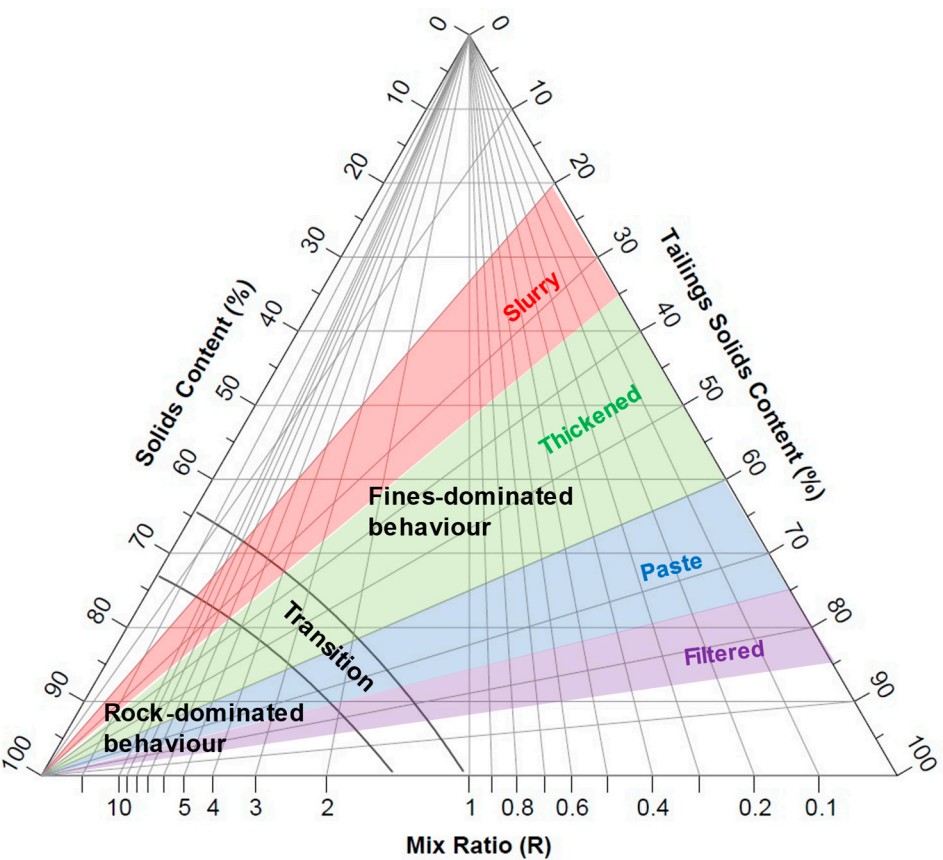

**Figure 7.** Ternary diagram showing blend configuration vs. mix ratio for different tailings dewatering technologies.

### 3.3. Commingled Filtered Tailings and Waste Rock

Established mix design theory has shown that the geotechnical behavior of a blend is governed by particle packing configuration, and that the configuration can be predicted from the mix ratio. However, the established theory is not applicable in the case of filtered tailings and waste rock blends that are placed loose and compacted under self-weight. This section presents an extended theoretical model to predict the structure and behavior of blends of waste rock and tailings, based upon mix ratio and density. It is based upon the co-disposal mix design theory developed by Wickland et al. [7], but is extended to consider the case of loosely placed materials and the effect of volume change on structural configuration.

When in an uncompacted state, filtered tailings and waste rock blends consist of discrete "lumps" of filter cake, rock particles and macro-scale (inter-lump) air voids. The lumps typically consist of compact-to-dense fines which are generally saturated or close to saturated at the time of deposition. In general terms, the geotechnical behavior of the blend is controlled by the structural configuration of the blend. The structural configuration is dependent upon the mix ratio and the packing density. Figure 8 presents a theoretical framework for describing the behavior of commingled filtered tailings and rock. The structural configurations and key processes that govern volume change under compression are shown in Figure 8.

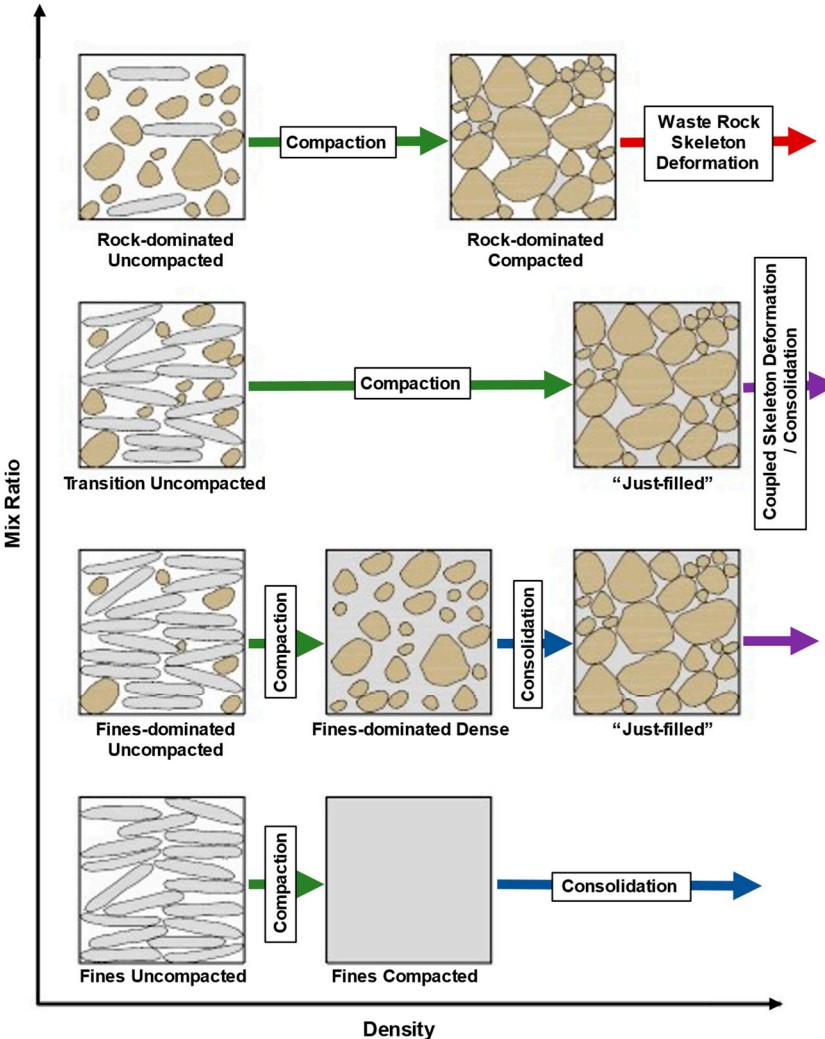

**Figure 8.** Structure and compression behavior of filtered tailings–waste rock blends.

In summary, five key structural configurations are proposed:

- Rock-dominated uncompacted;
- Fines-dominated uncompacted;
- Rock-dominated compacted;
- Fines-dominated compacted;
- "Just-filled".

"Just-filled" refers to a condition where a continuous, load-bearing waste rock "skeleton" exists, with the void spaces between the rock particles fully occupied with tailings. Figure 9 shows just-filled commingled tailings and waste rock at Porgera Mine in Papua New Guinea. "Fines-dominated compacted" refers to a condition where a discontinuous waste rock phase is "floating" in a continuous tailings matrix. "Rock-dominated compacted" refers to a condition with a continuous rock "skeleton", where tailings and air voids take up the void space between rock particles. Uncompacted blends (rock or fines dominated) consist of lumps of filter cake, rock particles and air voids.

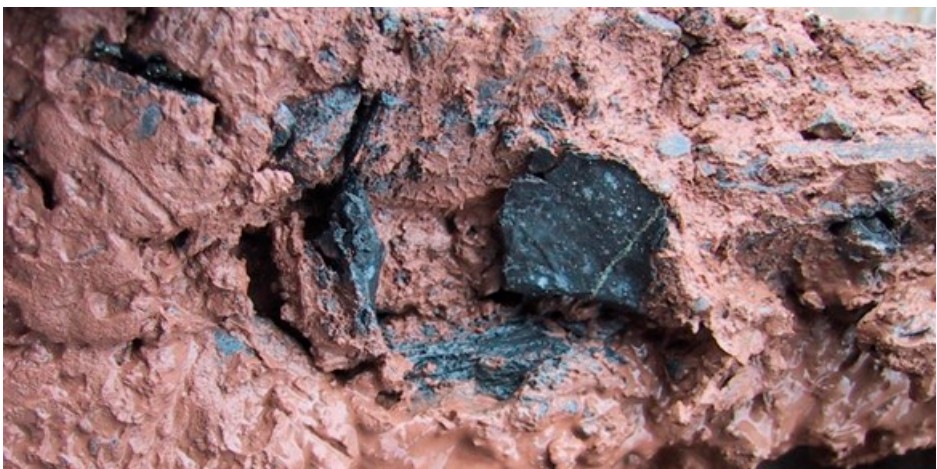

**Figure 9.** Example of "just-filled" commingled tailings and waste rock at Porgera Mine.

Four distinct processes that govern volume change under compression are proposed:

- Compaction (shown as green arrows in Figure 8);
- Waste rock skeleton deformation, or deformation of a continuous waste rock phase (shown as red arrows in Figure 8);
- Tailings consolidation, or deformation of a continuous tailings phase (shown as blue arrows in Figure 8);
- Coupled consolidation and skeleton deformation (shown as purple arrows in Figure 8).

Compaction refers to the process of removing large, inter-lump air voids by the deformation and breakdown of the filter cake lumps. Skeleton deformation refers to the deformation of the load-bearing waste rock skeleton after filter cakes are de-structured. Waste rock deformation occurs due to the rearrangement of the waste rock particle structure and particle breakage. Consolidation refers to the gradual reduction of the volume within the tailings fraction of the blend, due to drainage of pore water caused by an increase in pore pressure due to applied stress. Once the blend is in the "just-filled" condition, i.e., a continuous waste rock skeleton with the voids fully occupied by fine tailings, the compression behavior is governed by both consolidation and skeleton deformation.

Material property boundaries can be defined based on the initial tailings solids content and minimum void ratio of the waste rock skeleton, shown below in Figure 10.

Figure 10 shows the full range of possible conditions for a typical blend of waste rock and tailings with $0 \leq R \leq 5$. The chart defines the structural configuration and dominant volume change mechanism of a blend of a given mix ratio and density. A chart can quickly be generated for blends of specific materials based on easily obtainable properties. Inputs to the chart are the specific gravity of tailings solids and rock solids, the initial moisture contents of tailings and waste rock and maximum waste rock skeleton porosity.

A blend of a given mix ratio will trace a path horizontally across the chart from its initial dry density as it compresses. The blue line (tailings consolidation line) represents the point at which the blend becomes fully saturated, positive pore pressures are generated and the tailings start to consolidate. The position of this line is controlled primarily by the tailings solids content at deposition. The red line (waste rock skeleton deformation line) represents the point at which a continuous, load-bearing skeleton of waste rock is formed and deformation is governed by the waste rock skeleton. The position of this line is controlled primarily by the maximum void ratio of the waste rock skeleton. Once the blend has crossed both lines, deformation is governed by a coupled process involving both skeleton deformation and consolidation.

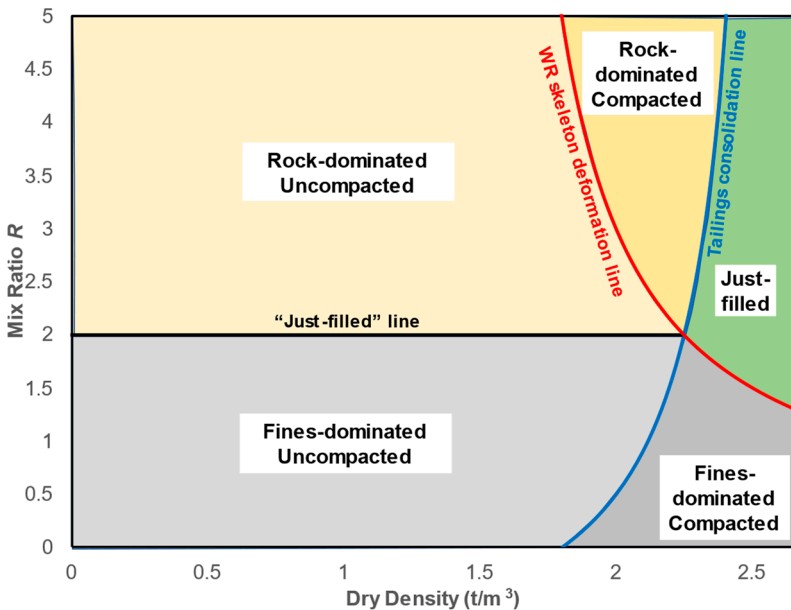

**Figure 10.** Material property boundary chart for a typical filtered tailings–waste rock blend.

The model proposed here is intended for mixtures where the waste rock has a low fines content and low moisture content, and the tailings are saturated and predominately fine. This is typical for run-of-mine waste rock blended with tailings in a hard rock, metal mining operation. The model presented here is not considered to be applicable for cases where the waste rock has a high fines content, high moisture content, high clay content or may be considered a very weak, soil-like rock that is prone to weathering. Considerations relating to these cases, for example in an oil sands operation, are discussed in the following section.

*3.4. Considerations Relating to Shales and Other Weak Rock Types*

In general, the mix design theories described in this paper are not applicable to blends composed of clay shale overburden or other highly weatherable, or soil-like, rock types. In contrast to waste rock particles, shale lumps are highly compressible due to the deformation and breakdown of the lumps. Consequently, it is difficult to predict the maximum void ratio of the shale lump skeleton. Furthermore, moisture is transferred from the tailings to the unsaturated shale lumps, causing them to swell. The internal porosity of the shale lumps must be considered.

At a high level, the blends can be characterized by final blend solids content, as the relationship between solids content and mix ratio can be easily calculated. A recommended method to predict blend structure and density from mix ratio is to do so by using experimental compaction tests, as described in the following section.

*3.5. Experimental Methods*

This section provides a brief discussion of experimental methods that may be used for the evaluation of commingled mine wastes.

The Proctor test is widely used in geotechnical practice to evaluate the compaction properties of soils, and provides a quick and effective method to evaluate the density of commingled materials at a range of mix ratios. The Proctor test may be effective in assessing the properties of blends of tailings and soil-like overburden materials. Figure 11 shows an example of a series of standard and modified Proctor tests carried out on oil sands tailings and clay shale overburden, prepared to a range of mix ratios. The plot is annotated with a line representing fully saturated conditions and a "just-filled" line representing 100% saturation of the inter-lump void space. This line can easily be calculated if the internal shale lump void ratio is known, which can be measured using standard test methods on intact lumps. In general, mixes would not be expected to plot above this line irrespective of

the degree of compaction. In this case, it can be seen that the optimum mix ratio (by dry mass) is around 6:1 shale solids–tailings solids, corresponding to a final solids content of approximately 75% by mass.

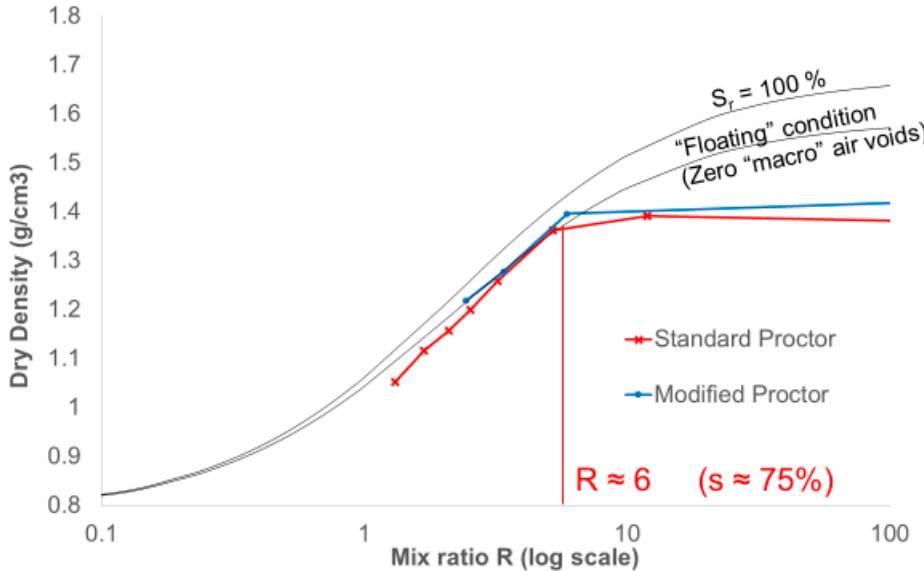

**Figure 11.** Dry density with respect to mix ratio for oil sands tailings–clay shale blends compacted using the standard and modified Proctor method.

An emerging technology with potential for wide-ranging application to the study of commingled mine wastes is X-ray computed tomography (CT) scanning. CT scanning is widely used in medicine, and its effectiveness has also been demonstrated in geotechnical applications, including the study of shear deformation and fracture features, as well as void ratio evolution during tri-axial testing [33–35]. A CT scanner is capable of measuring X-ray attenuation values for volumetric elements (known as "voxels") within its field of view, effectively creating a high-resolution 3D model of attenuation for the specimen. Attenuation can be linearly correlated with bulk density for the range of X-ray energies typically used in geotechnical applications [36].

Consequently, with calibration, a CT scanner and image processing software can be used to produce a 3D model of a sample of commingled mine wastes, which can be used to measure the volumes of constituent parts and visually evaluate particle structure, blend configuration and degree of mixing. Figure 12 shows an example of a CT scan carried out on a blend of commingled oil sands tailings and clay shale overburden. Figure 12a shows tailings in the voids of the commingled blend and Figure 12b shows shale overburden.

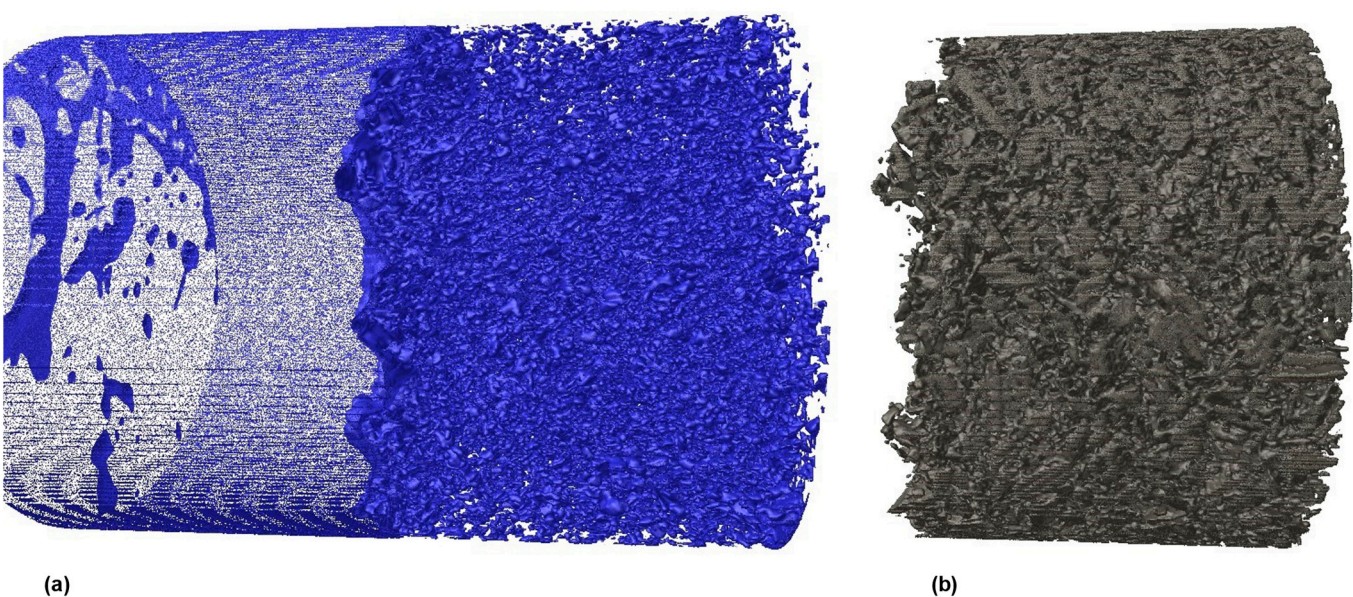

(a)                                                                                                          (b)

**Figure 12.** X-ray computed tomography scan of a sample of blended oil sands tailings and clay shale overburden, showing (**a**) tailings and (**b**) clay shale overburden.

## 4. Discussion and Conclusions

Commingling of waste rock and filtered tailings is an emerging technology that has advantages over traditional methods of mine waste disposal. Filtered tailings "dry stacking" is becoming an increasingly widely used alternative to traditional slurry tailings. The addition of waste rock to a filtered tailings stack has the potential to improve the stability of the deposit, possibly allowing for rapid stacking without compaction and for the dry stacking to be economical at the biggest mines.

Work by previous researchers, focusing primarily on blends of waste rock and paste tailings, has shown that the geotechnical behavior of a blend is governed by particle packing configuration and that the configuration can be predicted from the mix ratio. However, the established theory is not applicable in the case of filtered tailings and waste rock blends that are placed loose and compacted under self-weight. This paper presents an extended theoretical model to predict the structure and behavior of blends of waste rock and tailings, based upon mix ratio and density. It is based upon the co-disposal mix design theory developed by Wickland et al. [7] but is extended to consider the case of loosely placed materials and the effect of volume change on structural configuration. The extended model may be used to describe the geotechnical behavior of commingled tailings and waste rock, based upon the particle packing arrangement of the tailings. It is postulated that the geotechnical behavior of blends, and the primary mechanism of volume change, are governed by particle configuration. The model may be used to illustrate the structure and behavior of the blend, based on the mix ratio and density. The model may also be used as a framework for further investigations and the development of constitutive models to characterize these materials.

A methodology for the application of ternary diagrams to distinguish blends of waste rock and tailings blends is presented. These diagrams provide a useful method for quickly evaluating the properties of a blend of a given mix ratio and tailings solids content, which may assist in high-level tailings planning and design.

Specific conclusions of this paper include:

- Blends of commingled waste rock and filtered tailings can be characterized using five distinct structural configurations.
- The structural configuration of a specific blend is primarily dependent on the mixture ratio and the density of the blend. The configuration can be predicted based on these and other basic properties.

- It is postulated that the structural configuration of the blend defines the geotechnical behavior of the blend in general terms, specifically the mechanism of volume change in response to an increase in vertical stress. Four distinct mechanisms are proposed.
- Experimental methods such as CT scanning have the potential to allow the direct observation and measurement of blend structural configuration.

Recommendations for further study include the development of a rigorous mathematical model to characterize the behavior of filtered tailings and waste rock blends. The model would be required to model compaction, consolidation and waste rock skeleton deformation processes, and would have to consider both saturated and unsaturated conditions. The theoretical framework developed in this paper could be used as the basis for the model. Field-scale trials should be undertaken to study the stacking behavior of filtered tailings and waste rock blends to verify the experimental findings.

**Author Contributions:** Development of original theory, R.B.; experimental work, R.B.; writing—original draft preparation, R.B.; writing—review and editing, G.W.W.; project administration, G.W.W.; supervision, G.W.W. All authors have read and agreed to the published version of the manuscript.

**Funding:** This research was funded through the NSERC/COSIA Industrial Research Chair in Oil Sands Tailings Geotechnique, grant number IRCPJ 460863-18.

**Data Availability Statement:** Not applicable.

**Acknowledgments:** The authors would like to acknowledge the support of Mike London and Alberta Innovates for performing the CT scan and producing the image (Figure 12) presented in this paper.

**Conflicts of Interest:** The authors declare no conflict of interest. The funders had no role in the design of the study; in the collection, analysis, or interpretation of data; in the writing of the manuscript; or in the decision to publish the results.

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
