# Peer review of "Commingling of Waste Rock and Tailings to Improve “Dry Stack” Performance: Design and Evaluation of Mixtures"

_minerals, doi:10.3390/min13020295_

Round 1

Reviewer 1 Report

See attached pdf

Author Response

Please see attached fiel

Reviewer 2 Report

·        The organization of the article is not correct ( it must contain four sections). Sections are not ordered correctly. Please see  https://www.mdpi.com/journal/minerals/instructions.

·        Abstract is adequate to article content

·        Keywords are correctly proposed.

· References must be ordered according to applied science journal style; please see ( https://www.mdpi.com/journal/minerals/instructions).

·        Literature review is based on 15 references which are very low. References must be increased in order to figure out the gap.

· Materials and Methods sections must be presented

· The results in the article are well presented. This is a strong element of the article, but there is a lack of discussion.  Pay more attention to the discussion and it is important to make a comparison with previous relevant studies if any. I recommend that discussion section be separated from results.

· Please extend the conclusions part with future research direction, rewrite it in points.

Reviewer 3 Report

Figure 4: There is no reference to, or discussion of, Figure 4 in the text.  Additionally, it is not clear what is being measured on the vertical and horizontal axes of Figure 4.

Line 154: The section numbering goes from 3.1 directly to 3.2.1.  There appears to be a Section 3.2 heading, or perhaps a section itself, missing.

Section 3.2.4: The Figures in this Section need to be renumbered to conform with the numbering of the other Figures in the paper.

An issue I hope you can take up, perhaps in a future paper, is that of ensuring the stability of filtered tailings facilities.  What I often see these days is filtered tailings impoundments consisting of a shell of compacted tailings, with the interior tailings uncompacted.  This is essentially an upstream-type design. 

Unfortunately, some jurisdictions, for example Montana, have exempted filtered tailings facilities from tailings dam regulations.  I believe this will eventually lead to an unnecessary failure. For me, a filtered tailings impoundment is an engineered structure holding mine tailings.  This is the definition of a tailings dam.   

Any observations you might offer on how filtered tailings facilities should be designed to be as safe as possible, how they should be modeled to confirm their structural integrity, and what monitoring should be employed, would be helpful guidance.

Author Response

Errors in Figure references, figure numbering and  section numbering have been corrected.

Round 2

Reviewer 2 Report

  • There is still a weakness in the literature of the study, and the gap is not clear until now.
  • Also hoping that you  rephrase the conclusion and make it in points while providing it with significant results.
  • I will leave the issue of the structure of the paper to the editor, who in turn will decide whether it is appropriate.
  • Comparison with previous studies in the discussion should be stated well.

Author Response

bullet points summarising the specific conclusions of the have been added to the last chapter

minor edits / spelling corrections